# Probiotic Properties of *Pediococcus pentosaceus* JBCC 106 and Its Lactic Acid Fermentation on Broccoli Juice

**DOI:** 10.3390/microorganisms11081920

**Published:** 2023-07-28

**Authors:** Sang-Kyu Park, Hao Jin, Nho-Eul Song, Sang-Ho Baik

**Affiliations:** Department of Food Science and Human Nutrition, Jeonbuk National University, Jeonju 54896, Republic of Korea; dave0340@naver.com (S.-K.P.); blueberry-73jinhao@naver.com (H.J.); nesong@kfri.re.kr (N.-E.S.)

**Keywords:** *Jangajji*, lactic acid bacteria, cinnamoyl esterase, probiotic, immunomodulatory, broccoli fermentation

## Abstract

To understand the biological roles of *Pediococcus pentosaceus* strains as probiotics isolated from the traditional Korean fermented food, *Jangajji*, *Pediococcus pentosaceus* was selected based on its high cinnamoyl esterase (CE) and antioxidant activities. The acid and bile stability, intestinal adhesion, antagonistic activity against human pathogens, cholesterol-lowering effects, and immune system stimulation without inflammatory effects were evaluated. Nitric oxide (NO) levels were measured in co-culture with various bacterial stimulants. Fermentation ability was measured by using a broccoli matrix and the sulforaphane levels were measured. Resistance to acidic and bilious conditions and 8% adherence to Caco-2 cells were observed. Cholesterol levels were lowered by 51% by assimilation. Moreover, these strains exhibited immunomodulatory properties with induction of macrophage TNF-α and IL-6 and had microstatic effects on various pathogens. Co-culture with various bacterial stimulants resulted in increased NO production. Fermentation activity was increased with the strains, and higher sulforaphane levels were observed. Therefore, in the future, the applicability of the selected strain to broccoli matrix-based fermented functional foods should be confirmed.

## 1. Introduction

*Pediococcus pentosaceus*, a member of the genus *Latilactobacillus* (phylum Firmicutes, class Bacilli) and a representative homofermentative Gram-positive strain, is an emerging probiotic candidate. It has attracted much attention due to pediocin that can be used in food bio-preservation as well as its anti-inflammation [1,2], anticancer [3,4,5], antioxidant [6,7,8], detoxification [9,10], and lipid-lowering abilities [11,12]. To date, many strains have been isolated from fermented foods [13,14], animals [15], plants [16], and fecal sources [17]; however, there is still insufficient information and a lack of knowledge regarding their probiotic and fermentative properties. Although *P. pentosaceus* and its pediocin are generally considered safe for use in foods (due to their ‘generally recognized as safe (GRAS) status), it is essential to evaluate their probiotic safety aspects in the human intestine, such as antimicrobial resistance and toxicological safety, through cytotoxicity assays [18,19].

Broccoli (*Brassica oleracea* L. var. *Italica Plenk*) is a vegetable belonging to the genus Brassica (family Brassicaceae or Cruciferae) and includes cauliflower, Brussels sprouts, kohlrabi, cabbage, and mustard [20]. It is an important source of various nutrients, including dietary fiber, vitamins, minerals, and phytochemicals such as glucosinolates and phenolic compounds. Recently, interest in broccoli and its bioactive compounds has increased, as many studies have demonstrated the beneficial effects of broccoli glucosinolates and their breakdown products, particularly sulforaphane, on antioxidant, anti-inflammatory, anti-cancer, metabolic syndrome, and neuroprotective properties [21]. The most abundant glucosinolate in broccoli is glucoraphanin [22], which can be hydrolyzed by the endogenous enzyme myrosinase when broccoli is cut or chewed to form isothiocyanate, sulforaphane, a potent anti-cancer substance that works by increasing the level of enzymes in the liver [23]. Based on research that improved the yield and stability of sulforaphane in broccoli and increased the potential health benefits by improving the antioxidant ability through *Lactobacillus* fermentation [24], broccoli was selected as a raw material for examining the effectiveness of the selected *P. pentosaceus* strain, because it is likely to be a functional fermented food preparation containing various antioxidants and anticancer substances.

The aim of this study was to evaluate the probiotic properties of an isolate of *P. pentosaceus* JBCC 106 originating from the traditional Korean fermented food, Jangajji, and its viability and functionality in broccoli juice.

## 2. Materials and Methods

### 2.1. Lactic Acid Bacteria Strain Isolated from Jangajji

Lactic acid bacteria (LAB) were isolated from the traditional Korean fermented pickle, *Jangajji*, a high caffeic acid producer, by Baik et al. (2022, patent no. KACC92458P). The strains were routinely prepared in MRS broth (Difco Laboratories, Detroit, MI, USA) and grown at 37 °C for 24 h. *L. rhamnosus* GG (ATCC 53103, LGG), a representative probiotic strain, was used as the reference control to evaluate the probiotic properties of the isolated strains.

### 2.2. Identification of the Isolated Strains

16S rDNA sequencing was performed as described previously [25]. Sequence homology analysis was performed by comparing the obtained sequences with LAB sequences in the DNA database (http://www.ncbi.nlm.nih.gov/BLAST, accessed on 13 April 2022).

### 2.3. CE Activity

CE activity was measured quantitatively and qualitatively using thin-layer chromatography (TLC) and high-performance liquid chromatography (HPLC) as previously described [26]. LAB strains (1%, *v*/*v*) were inoculated with 5 mL of MRS broth and incubated at 37 °C for 72 h in the presence of a 0.5% (*w*/*v*) chlorogenic acid (Sigma, St. Louis, MO, USA). Subsequently, 0.1% (*w*/*v*) ascorbic acid was added to the culture and extracted with 2.5 mL ethyl acetate. TLC was performed on silica gel 60 F254 plates (Merck, Darmstadt, Germany) using toluene/ethyl acetate/formic acid (5:4:1, *v*/*v*/*v*) as the solvent system, as described by Kim et al. [26]. HPLC analysis was performed on an Accela (Thermo Scientific, Waltham, MA, USA) equipped with an Inertsil ODS-3 V column (150 × 4.0 mm i.d., 5 m). The mobile phase was 0.5% formic acid and acetonitrile (*v*/*v*, 75:25) at a flow rate of 1.0 mL for 10 min at 40 °C. Each sample was analyzed in triplicate.

### 2.4. Antioxidant Activity

Antioxidant activity was measured by 2,2′-Azinobis (3-ethylbenzothiazoline-6-sulfonic acid) (ABTS) radical scavenging activity, α-diphenyl-β-picrylhydrazyl (DPPH) radical scavenging activity, and superoxide dismutase (SOD) activity as follows. ABTS radical scavenging activity was determined as described by Lee et al. [27], with some modifications. The ABTS solution (1 mL) was mixed with 3 mL LAB samples and reacted for six minutes. After the reaction, absorbance was measured at 734 nm. The DPPH radical-scavenging activity of the LAB was determined according to the method described by Li et al. [28]. The SOD activity of the LAB was measured spectrophotometrically according to the method described by Song et al. [29] with some modifications. Briefly, the culture supernatant of LABs was mixed with reaction solution (0.5 mM, Tris-HCl buffer, pH 8.2, and 7.2 mM pyrogallol) at the ratio of 1:3:0.2 and incubated for 10 min at 25 °C. After incubation, the reaction was terminated by adding 1 N HCl (0.5 mL) and the absorbance was measured at 420 nm.

### 2.5. Probiotic Properties

#### 2.5.1. Acid Resistance and Bile Tolerance

The acid resistance of the selected LAB was determined according to the method described by Liong and Shah [30] with some modifications. After cultivation in MRS broth at 37 °C, the recovered cell was resuspended in an equal volume of MRS broth (final pH 1.5 and 2.0) and incubated at 37 °C for 2 h. For bile tolerance, 100 μL culture was dropped onto MRS agar plates containing 0%, 0.5%, 1%, and 3% (*w*/*v*) of bile (Sigma Chemical Co., St. Louis, MO, USA), and incubated at 37 °C for 24–48 h [31]. Acid and bile tolerance were determined based on the survival rate. The survival rate was calculated according to the following:(1)Survival rate (%): (log A1/A0)×100% .

*A*1: Total viable count of probiotic strains after treatment, *A*0: Total viable count of probiotic strains before treatment.

#### 2.5.2. Bile Salt Hydrolase (BSH) Activity

Sterile discs were placed on a BSH screening medium, and 100 μL of the selected LAB strain culture was added. BSH screening medium was prepared using MRS agar supplemented with 0.5% (*w*/*v*) sodium salt, taurodeoxycholic acid (Sigma Aldrich, St Louis, MO, USA), and 0.37 g/L of CaCl_2_ [32].

#### 2.5.3. Antimicrobial Activity

Antimicrobial activity was determined using a paper disc according to the method described by Chiu et al. [33] with minor modifications. A total of 50 μL of supernatants of LAB strain supernatant was dropped onto the discs on nutrient broth agar plates. Each indicator pathogenic strain was grown in nutrient broth at 37 °C for 24 h and sterile 8 mm paper discs with LABs were placed on the agar. After the plates were incubated for 24 h at 37 °C, the diameter of the clear zone was measured.

#### 2.5.4. Antibiotic Resistance

Ampicillin and vancomycin as inhibitors of cell wall synthesis and kanamycin and chloramphenicol as inhibitors of protein synthesis (Sigma) were used. All antibiotic powders were dissolved in appropriate diluents and filter-sterilized before addition to MRS medium. To test antibiotic resistance, LABs were inoculated (1%, *v*/*v*) in MRS broth supplemented with antibiotics at various final concentrations (from 2 to 1024 mg/mL) and measured the growth at 580 nm after 24 h incubation at 37 °C. The MIC was defined as the lowest concentration of antibiotics that completely inhibited visible growth in comparison with an antibiotic-free control well.

#### 2.5.5. Cholesterol Assimilation

The ability of lactobacilli to assimilate cholesterol was measured according to the methods described by Ren et al. [34] and Liong et al. [30]. Cholesterol assimilation was calculated using the following equation:(2)Cholesterol assimilation (%)=(1−A1/A0)×100

*A*1: Cholesterol presents MRS at time 0 h, *A*0: Cholesterol presents MRS at time 24 h.

#### 2.5.6. Bacterial Adhesion Capacity

##### Cell Surface Hydrophobicity

Bacterial adhesion to hydrocarbons was determined following the method of Kaushik et al. [35] Surface hydrophobicity was calculated as the percentage decrease in the absorbance of the aqueous phase after mixing and phase separation relative to that of the original suspension (AbsInitial) using the following equation:(3)Surface Hydrophobicity (%)=100×(AbsInitial−Absfinal) / Absinintial △ Abs/Absinintial×100

##### Cell Aggregation

The auto-aggregation assay was performed according to the method described by Kaushik et al. [35]. The percentage difference between the initial and final absorbances provides an index of cellular auto-aggregation, which can be expressed by the following equation:(4)Aggregation (%)=100×(AbsInitial−Absfinal)/AbsInitial
(5) =△ Abs/AbsInitial×100

##### Probiotic Adhesion to Caco-2 Cells

An in vitro adhesion assay was performed using the method described by Ren et al. [34]. Adhesion to Caco-2 cells was calculated using the following equation:(6)Adhesion to Caco−2 cells (%)=100×(Absinintial−Absfinal) / Absinintial
(7)=△ Abs/Absinintial×100

### 2.6. Cell Cytotoxicity

Cytotoxicity was assessed using the methylthiazolyltetrazolium (MTT) assay, as described by Levy and Simon [36]. RAW 264.7 cells were seeded at a density of 5 × 10^4^ CFU/mL in 96-well plates (Nunc, Roskilde, Denmark) for 24 h. Absorbance at 450 nm was measured using a microplate spectrophotometer. The results were expressed as absorbance values measured at 450 nm with the respective controls.

### 2.7. NO and Cytokine (TNF-α and IL-6) Measurement

NO production was determined based on the level of nitrite formed in the supernatant of the cultured RAW 264.7 cells. The cells were seeded at 5 × 10^4^ CFU/mL on 96-well culture plates for 2 h. Either viable or heat-inactivated bacteria at representing bacteria (100 μL) were added to the wells. After 24 h incubation (37 °C, 5% CO_2_), the supernatant was collected and NO and cytokines were measured. NO production was measured using the Griess reaction [37].

### 2.8. Lactic Acid Fermentation on Broccoli Juice

Broccoli (*Brassica oleracea* var. *italica*) used in this experiment was purchased from a local market in Jeonju, Korea. Broccoli juice was prepared as described by Xu et al. [38] with several modifications. After washing the broccoli with distilled water, the leaves, and stalks were removed, and the florets were homogenized in sterile distilled water at a ratio of 1:2 (broccoli/distilled water, g/mL). The prepared broccoli juice was filtered through a cloth to separate the solid components, transferred to an Erlenmeyer flask, and pasteurized at 60 °C for 5 min. *P. pentosaceus* JBCC 106 was pre-cultured in MRS broth medium at 37 °C for 24 h and inoculated into the broccoli juice and fermented at 37 °C for 36 h. During fermentation, 30 mL samples were collected at 0, 3, 6, 12, 24, and 36 h and stored at −20 °C until analysis. The pH of the stored samples was determined using a pH meter (Starter3100, Ohaus, Parsippany, NS, USA) and viable cell count was measured by dispensing each sample diluted in sterilized distilled water onto an MRS agar plate and incubating at 37 °C for 24 h. All experiments were conducted in triplicate.

### 2.9. Analysis of Sulforaphane from Broccoli Juice

Sulforaphane was extracted from fermented broccoli samples as described by Ghawi et al. [39] and Abukhabta et al. [40] with some modifications. A total of 2 mL of the sample was placed in a tube and incubated for 5 h at 30 °C to ensure complete hydrolysis of glucosinolates by myrosinase. Samples were then centrifuged at 13,000× *g* for 10 min, the pellet was dissolved in 2 mL of distilled water, and the previous process was repeated. To extract sulforaphane from the supernatant, 10 mL of dichloromethane was added to the collected supernatant in a conical tube, vortexed for 15 min, and centrifuged at 13,000× *g* for 10 min. The organic phase was then collected, and 10 mL of dichloromethane was added to the pellet for re-extraction. The collected organic phase was concentrated using a rotary evaporator (EYELA N-1110, Tokyo, Japan) and suspended in 6 mL of acetonitrile to prepare a sample for HPLC analysis.

Sulforaphane levels were analyzed by HPLC (Thermo Fisher Scientific Inc., Waltham, MA, USA), which consisted of an accela 600 pump (San Jose, CA, USA) equipped with an accela PDA 80 hz detector (San Jose, CA, USA) using an Exsil 100 ODS column (250 × 4.6 mm, 5 µm; Exmere LTD., Lancashire, UK). The sample was filtered through a 0.45 μm membrane before analysis. The mobile phase was acetonitrile/water (30:70) at a flow rate of 0.6 mL/min and the injection volume was 10 µL.

### 2.10. Statistical Analysis

All experiments were performed in triplicate. The error bars in the graphs indicate standard deviation. Data were analyzed using analysis of variance (ANOVA) and Student’s *t*-test. Differences were considered statistically significant at *p* < 0.05.

## 3. Results and Discussion

### 3.1. Bacterial Strains

The LAB used in this study were isolated from five traditional Korean fermented foods, *Jangajji* and were identified as *P. pentosaceus* JBCC 106 using full-length 16S rDNA sequences. (GenBank accession number: NR_042058; Appendix A).

### 3.2. Physicochemical Properties of P. pentosaceus JBCC 106

When this isolate was examined for in vitro CE activity using chlorogenic acid as substrate, CA was produced up to 30 µM with a conversion ratio of 16.4% (Table 1), which was slightly lower than that of previous LAB strains [41]. To the best of our knowledge, there have been few studies on *P. pentosaceus* related to CE activity. The antioxidant capacity of the isolated strain was explained through SOD activity, ABTS, and DPPH radical scavenging activities, which were 39%, 41%, and 45%, respectively, which had better antioxidant abilities than strains isolated from *Jangajji* and *Jeotgal* in other studies [42].

### 3.3. Probiotic Properties of P. pentosaceus JBCC 106

#### 3.3.1. Resistance in Artificial Gastrointestinal Conditions

As shown in Table 2, *P. pentosaceus* JBCC 106 exhibited moderate survival of 79.4% at pH 3.0, which was lower survivability than *L. rhamnosus* GG, but showed higher acid tolerance at pH 2.0. *P. pentosaceus* JBCC 106 was tolerant to even 3% (*w*/*v*) oxgall, as evidenced by survival rates greater than 97% even after 24 h exposure (Table 2). However, *P. pentosaceus* JBCC 106 showed moderate BSH activity but a good survival rate at a reduced pH of 1.5. Bile salt tolerance is closely related to both BSH and osmotic stress tolerance. Because BSH activity is not common among bacteria isolated in non-bilious environments, such as cheese or fermented vegetables [43,44,45], it is not surprising that most of the isolates showed low BSH activity. We measured BSH activity in strains treated with simulated gastric juice for 2 h because LAB is in a state of acid stress that must resist exposure to bile secreted into the small intestine. Although it is not clear whether BSH activity is a desirable trait for the selection of promising probiotic strains, our results suggest that this activity could maximize the intestinal survival of *P. pentosaceus* JBCC 106, increasing their overall beneficial effects related to *P. pentosaceus* JBCC 106 [46].

#### 3.3.2. Antimicrobial Activity and Antibiotic Susceptibility of *P. pentosaceus* JBCC 106

As shown in Table 2, *P. pentosaceus* JBCC 106 exhibited diverse antimicrobial activities against different Gram-positive and Gram-negative pathogens, such as *S. aureus*, *S. epidermidis*, and *S. xylosus*, and Gram-negative strains of *P. aeruginosa*, *P. putida*, *Pro. acnes*, *B. cereus*, and *B. vallismortis*, except *Escherichia coli. P. pentosaceus* has played an increasingly pivotal role in the LAB industry in recent years. Many strains of *P. pentosaceus* have been isolated from various sources, such as fermented food, aquatic products, raw animal meats, plant products, and feces. Additionally, links to the human gastrointestinal tract have finally been proven. This is in good agreement with *P. pentosaceus*, which generally produces a variety of functional compounds such as pediocin, exopolysaccharides, bacteriocin-like inhibitory substances (BLISs), and 3-phenyllactic acid, resulting in broad-spectrum antibacterial properties against a variety of food-spoiling bacteria and fungi. We also determined the MIC, which is the lowest antibiotic concentration that can inhibit bacterial growth [47]. MICs ≥ 8 μg/mL are considered “moderately resistant”; above 32 μg/mL is considered “clinically resistant” [48]. As shown in Table 2, *P. pentosaceus* JBCC 106 was resistant to both vancomycin and kanamycin with a significantly high MIC of 64 μg/mL. According to EFSA (2012), the resistance to vancomycin and kanamycin of *P. pentosaceus strains* and even *L. rhamnosus* GG as a control can be considered native non-transferable to other species because of embedded chromosomal genetic features which lack transportable properties [49].

#### 3.3.3. Cholesterol-Lowering Effect of *P. pentosaceus* JBCC 106

As shown in Figure 1, *P. pentosaceus* JBCC 106 showed no significant difference in cholesterol-lowering effects compared to the control strain *L. rhamnosus* GG (*p* > 0.05). It has been known that the presence of BSH activity in a probiotic strain has always been linked to cholesterol-lowering potential and crucial indicator for the selection of probiotic strain adjuncts to manage hypercholesterolemia. The presence of BSH in probiotics renders them more tolerant to bile salts, which helps reduce host blood cholesterol levels [50].

#### 3.3.4. Cell Surface Hydrophobicity, Auto-Aggregation, and Bacterial Adhesion to Caco-2 Cells of *P. pentosaceus* JBCC 106

Mucosal surface adhesion ability is another commonly encountered criterion for probiotic strain selection because it is directly related to colonization and persistence in the GI tract [51]. As shown in Figure 2A, heat-inactivated *P. pentosaceus* JBCC 106 cells showed an adhesion percentage of 8%, which was not significantly different from that of the reference probiotic strain of *L. rhamnosus* GG (*p* > 0.05). *P. pentosaceus* JBCC 106 exhibited a good hydrophobic cell surface, with a high percentage of 28%, but a low auto-aggregation rate of 18% (Figure 2B). *P. pentosaceus* JBCC 106 strain had higher hydrophobicity and auto-aggregation rates than the control *L. rhamnosus* GG.

### 3.4. Immunological Activity of P. pentosaceus JBCC 106

*P. pentosaceus* JBCC 106 showed no cytotoxic effects on macrophages at up to 8 and 9 when the OD values shown in Figure 3 were converted to log CFU/mL, indicating *P. pentosaceus* JBCC 106 did not cause cytotoxicity, indicating its toxicological safety for further application as a probiotic. Then, heat-killed whole bacterial cells of the isolated LAB strain were added to RAW 264.7 cell cultures, and NO production was evaluated to examine the anti-inflammatory activity of the LAB isolate in LPS-stimulated RAW 264.7 cells. NO has various biological functions in many types of immune cells, including induction of bactericidal effects in macrophages and signal transduction during inflammation [52]. Additionally, NO is the most important factor in vasodilation, which relaxes the inner muscles of blood vessels, causing them to dilate and increase circulation [53]. As shown in Figure 3B, *P. pentosaceus* JBCC 106 critically showed increased NO production in the RAW 264.7 cell cultures compared to the LPS induced control, indicating that *P. pentosaceus* JBCC 106 may be effective as a probiotic. We also examined cytokine induction activity, which contributes to maintaining inflammatory homeostasis in the gut mucosa under normal conditions. As shown in Figure 3C, the production of the pro-inflammatory cytokines TNF-α and IL-6 in macrophages treated with *P. pentosaceus* JBCC 106 was identical to that of the control strain, *L. rhamnosus GG*, which showed significantly reduced levels compared to LPS RAW 264.7 cells.

### 3.5. Fermentation of Broccoli Juice by Pediococcus pentosaceus JBCC 106

*P. pentosaceus* JBCC 106 was inoculated into pasteurized broccoli juice and fermented for 36 h to measure the viable cell counts and pH changes to evaluate the fermentation activity in the broccoli matrix. As shown in Figure 4A, *P. pentosaceus* JBCC 106 grew well from an initial live cell concentration of 6.4 Log CFU/mL to 7.5 Log CFU/mL until 12 h of fermentation and tended to decrease. The pH of the broccoli juice decreased sharply from 5.8 to 3.9 until 12 h, and then gradually decreased to 3.8. During fermentation, the antioxidant activity of the fermented broccoli juice was calculated using SOD, ABTS, and DPPH radical scavenging activities. The SOD activity of broccoli juice increased by about 60% compared to that of raw broccoli juice before fermentation, and the ABTS and DPPH radical scavenging activities increased by about 90% and 40% after fermentation, respectively (Figure 4B). We also examined sulforaphane content change during LAB fermentation on broccoli juice by *P. pentosaceus* JBCC 106. The sulforaphane content in broccoli juice increased approximately 3.2-fold after LAB fermentation by *P. pentosaceus* JBCC 106 (Figure 5, 31 g/mL), demonstrating that *P. pentosaceus* JBCC 106 can enhance antioxidant and sulforaphane contents when fermented as a starter in a broccoli matrix. However, further optimization of the method to enhance sulforaphane content is needed.

## 4. Conclusions

This study addressed the CE activity of *P. pentosaceus*, evaluated the probiotic potential of *P. pentosaceus* JBCC106, and evaluated its effect on broccoli fermentation using *L. rhamnosus* GG as a control. *L. rhamnosus* GG is a representative probiotic with excellent acid resistance and adhesion, and both proved to have similar probiotic properties. In addition, when the isolated strain was inoculated into a broccoli matrix as a starter and fermented, its applicability to fermentation was confirmed, with an increase to antioxidant activity and sulforaphane contents. These experimental results suggest that *P. pentosaceus* JBCC 106 is a good candidate for use as a probiotic supplement and has the potential to be applied in various functional fermented foods, including broccoli.

## Figures and Tables

**Figure 1 microorganisms-11-01920-f001:**
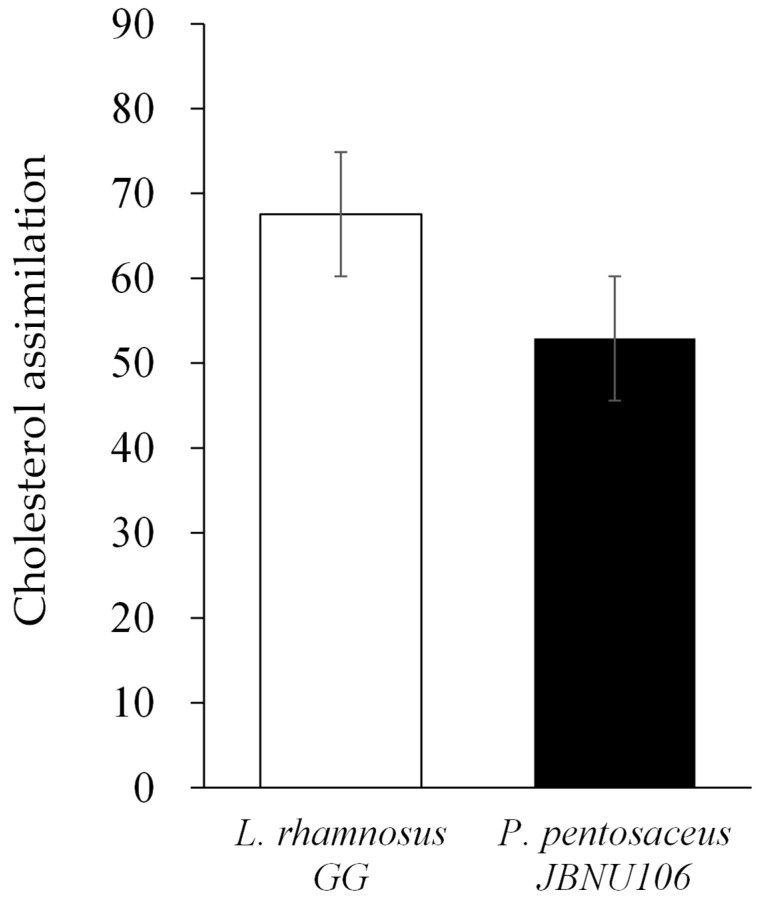
Comparison of the cholesterol-lowering ability between the strain isolated from *Jangajji* and the control strain, *L. rhamnosus* GG.

**Figure 2 microorganisms-11-01920-f002:**
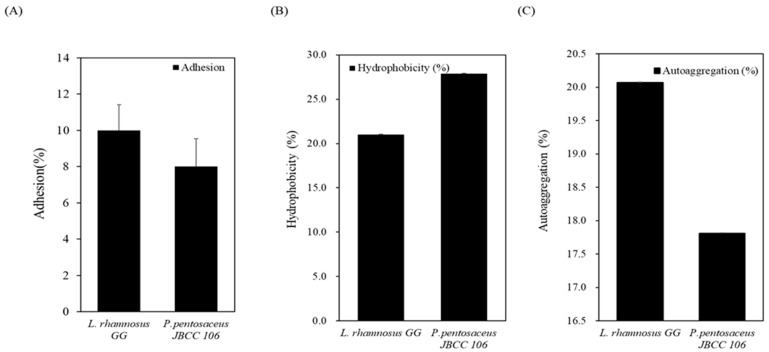
Mucosal surface attachment ability of the strain isolated from *Jangajji* and the control strain, (**A**) shows bacterial adhesion to Caco-2 cells, and (**B**) shows cell surface hydrophobicity and (**C**) shows auto-aggregation.

**Figure 3 microorganisms-11-01920-f003:**
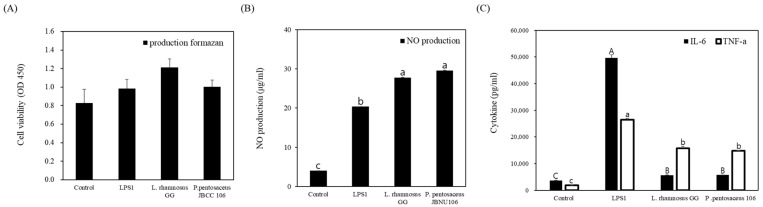
The immune-related activity of the strain isolated from *Jangajji* and the control strain. (**A**) shows the cytotoxic effect of the strain on 264.7 cells, (**B**,**C**) show the effect on anti-inflammatory indicators such as nitric oxide and cytokine secretion in 264.7 cells. Data are presented as the means for the three independent replicates (mean ± SD). Values that do not share the same letter are significantly different (*p* < 0.05).

**Figure 4 microorganisms-11-01920-f004:**
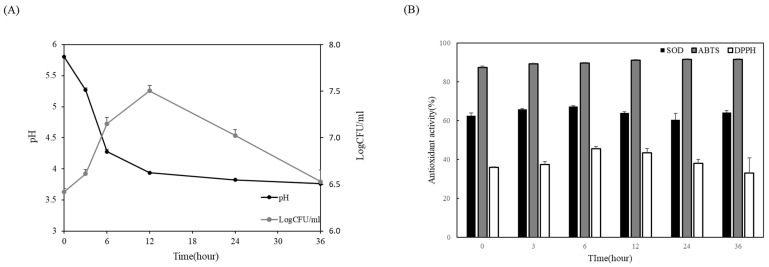
The strain isolated from *Jangajji* was applied to broccoli juice as a starter for fermentation, (**A**) shows the growth characteristics of the strain during the fermentation process, and (**B**) shows the antioxidant activity change of broccoli juice during fermentation time.

**Figure 5 microorganisms-11-01920-f005:**
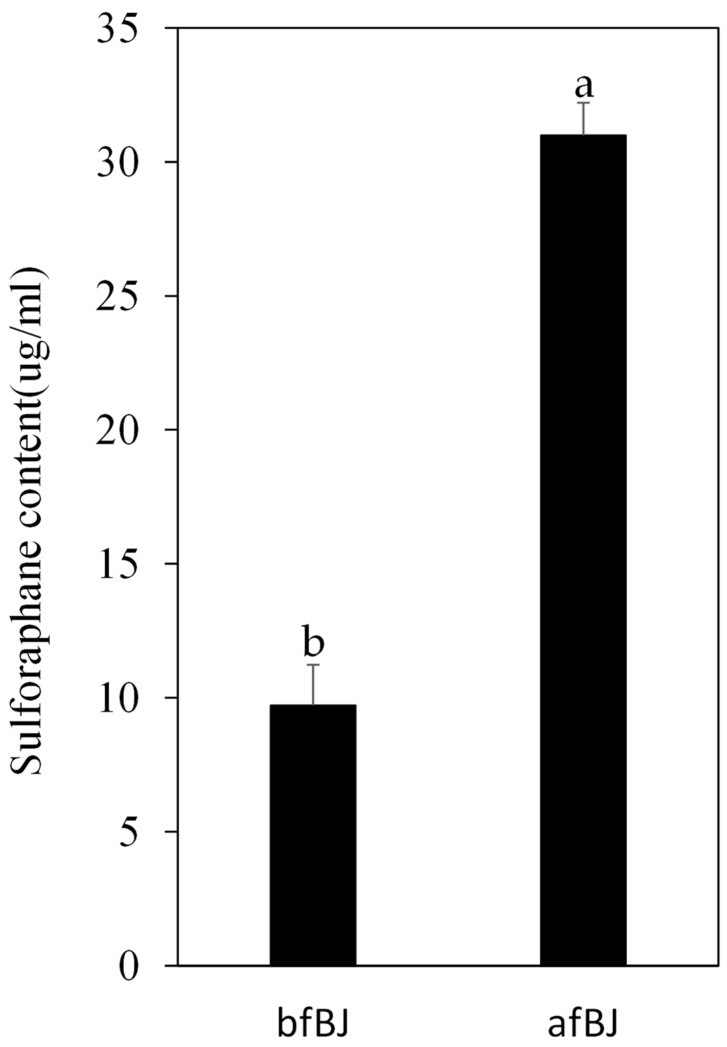
Changes in sulforaphane contents in broccoli juice fermentation using strains isolated from *Jangajji* as a starter. Different lower-case letters indicate that the values are significantly different (*p* < 0.05). bfBJ, broccoli juice before fermentation; afBJ, broccoli juice after fermentation.

**Table 1 microorganisms-11-01920-t001:** Identification and physicochemical properties of strains isolated from the traditional Korean fermented food, *Jangajji*.

LAB Isolates	16s rDNA Analysis		CE Activity		Antioxidant Activities (%)		
	Strains	Homology (%)	Generated CA (mM)	Conversion Rate (%)	ABTS	DPPH	SOD
JBCC 106	*Pediococcus* *pentosaceus*	99	0.03	16.39	41.1 ± 0.09	45.5 ± 0.18	±0.01

**Table 2 microorganisms-11-01920-t002:** Basic probiotic properties of the strain isolated from *Jangajji* and comparison with the control strain, *L. rhamnosus* GG.

Probiotic Properties		Strains	
		*L. rhamnosus* GG	*P. pentosaceus* JBCC 106
Acid tolerance (Survival rate, %)	pH 3	95.4	79.4
	pH 2	29.9	39.9
	pH 1.5	38.6	53.3
Bile salt tolerance (Survival rate, %)	0.5% bile salt	97.7	98
	1% bile salt	94.9	97.5
	3% bile salt	88.7	99.5
Pathogens ^a^	*S. aureus*	++	++++
	*S. epidermidis*	++	++
	*P. aeruginosa*	++	++
	*Pro. acnes*	-	+
	*P. putida*	++	±
	*S. xylosus*	+++++	++++
	*E. coli*	++	-
	*B. cereus*	+++	+
	*B. vallismortis*	++	++
MICs ^b^ (μg/mL)	Ampicillin (A)	<2	<2
	Chloramphenicol (C)	<2	4
	Vancomycin (V)	64	64
	Kanamycin (K)	≥1024 ^R^	64

^a^ Antimicrobial activity (zone size): - not clear zone; ± 0–1 mm; + 1–2 mm; ++ 2–3 mm; +++ 3–4 mm; ++++ 4–5 mm; +++++ >5 mm. ^b^ MIC: minimum inhibitory concentration. ^R^ Resistant according to the EFSA’s breakpoints (EFSA, 2012).

## Data Availability

Not applicable.

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
