# Peer review of "Probiotic Properties of Pediococcus pentosaceus JBCC 106 and Its Lactic Acid Fermentation on Broccoli Juice"

_microorganisms, 2023, doi:10.3390/microorganisms11081920_

Round 1
Reviewer 1 Report
The Ms (Probiotic Properties of Pediococcus pentosaceus JBCC 106 and its Lactic Acid Fermentation on Broccoli Juice 3) is interesting and have good data. Well designed and presented.
I have some minor comments.
The authors should highlight the cinnamoyl esterase in the title and not to focus on lactic acid fermentation as it missing data on broccoli fermentations. Also correct the formatting of Pediococcus pentosaceus
Expand NO level as first mention at L15
In the abstract show all utilized strains in this study as co-culture
L65 correct the format of Lactobacillus
L170-172, and 108 check for sub/superscript for 104 , CO2 , 37oc
L241/261/268 correct P. pentosaceus
L297 change the title to Fermentation of broccoli Juice by Pediococcus pentosaceus JBCC 106
Carefully check all scientific names of Microorganisms at reference section as all are incorrect format.
Author Response
July 20, 2023
Microorganisms
Manuscript number: microorganisms-2497741
Title: Probiotic Properties of Pediococcus pentosaceus JBCC 106 and its Lactic Acid Fermentation on Broccoli Juice
We thank the editor and all reviewers for their thorough and constructive comments. We have addressed each comment in detail below, and the changes made to the manuscript are highlighted in the text. We kindly request your reconsideration of our manuscript for publication in Microorganisms.
Reviewer 1
Q1) The authors should highlight the cinnamoyl esterase in the title and not to focus on lactic acid fermentation as it missing data on broccoli fermentations. Also correct the formatting of Pediococcus pentosaceus
R1) We thank the reviewer for providing valuable comments and suggestions that have helped us improve the quality of our submitted manuscript. Actually, our purpose in this study did not focused on CA, but probiotic properties of Pediococcus pentosaceus JBCC106 although this strain selected as high CA producer. Thus we changed our introduction by removing abundant explanation on CE activity to focus our study on probiotic properties of Pediococcus pentosaceus JBCC106. We also corrected the formatting of Pediococcus pentosaceus by adding italics. Thanks.
Q2) Expand NO level as first mention at L15
R2) We expanded NO to “Nitric oxide” in line 15. Thanks.
Q3) L65 correct the format of Lactobacillus
R3) Thanks. We changed it.
Q4) L170-172, and 108 check for sub/superscript for 104 , CO2 , 37oc
R4) Thanks. We changed them.
Q5) L241/261/268 correct P. pentosaceus
R5) Thanks. We added italics.
Q6) L297 change the title to Fermentation of broccoli Juice by Pediococcus pentosaceus JBCC 106
R6) We changed the title of L297 to what you suggested. Thanks.
Reviewer 2 Report
The authors attempted to evaluate the probiotic properties of P. pentosaceus JBCC 106 and its fermentation characteristics in broccoli juice. The innovation of the paper is average, but the experimental design is complete.
Detailed comments are as follows:
1. Since the authors emphasized the potential probiotic function of cinnamoyl esterase producing bacteria, why not screen for CA producing strains with greater capacity from multiple LAB strains. The authors need to explain the rationale for directly selecting P. pentosaceus JBCC 106 for testing its CA-producing ability in this study.
2. The authors used cinnamoyl esterase producing bacteria as an entry point, but why did they not test the ability of CA production in the later application of fermenting broccoli juice. After 36 h of fermentation of broccoli juice, the number of viable bacteria dropped sharply (figure 4A). If we put aside the fact that the antioxidant capacity of broccoli juice increased after fermentation (because the content of the antioxidant substance: sulforaphane increased significantly after fermentation of broccoli juice), does this application make sense? Since P. pentosaceus JBCC 106 has good antioxidant properties (table 1) and the fermentation of broccoli juice by this strain significantly increased the content of sulforaphane (an antioxidant substance, figure 5), why did the antioxidant capacity of broccoli juice not show a significant increase before and after fermentation (figure 4B), please The authors provide a reasonable explanation.
3. In lines 227-229 (Table 2), P. pentosaceus JBCC 106 is less tolerant than LCC at pH 3, but significantly better than Lactobacillus rhamnosus at pH 2 and 1.5, please give a reasonable explanation.
4. In line 162, I think it is more appropriate to use cell cytotoxicity than cell viability.
5. Some writing errors, such as lines 170, 172, 241, 261, 268.
Author Response
July 20, 2023
Microorganisms
Manuscript number: microorganisms-2497741
Title: Probiotic Properties of Pediococcus pentosaceus JBCC 106 and its Lactic Acid Fermentation on Broccoli Juice
We thank the editor and all reviewers for their thorough and constructive comments. We have addressed each comment in detail below, and the changes made to the manuscript are highlighted in the text. We kindly request your reconsideration of our manuscript for publication in Microorganisms.
Reviewer 2
Q1) Since the authors emphasized the potential probiotic function of cinnamoyl esterase producing bacteria, why not screen for CA producing strains with greater capacity from multiple LAB strains. The authors need to explain the rationale for directly selecting P. pentosaceus JBCC 106 for testing its CA-producing ability in this study.
R1). We selected the Pediococcus pentosaceus JBCC106 LABs from Jangajjii by CA producing ability as mentioned L 75 of Materials and Methods. However, our purpose in this study did not focused on CA, but probiotic properties of Pediococcus pentosaceus JBCC106 although this strain selected as high CA producer. Thus, we changed our introduction by removing abundant explanation on CE activity. Although several other strains in our culture collection showed more CA activity than our selected strain, but other strains did not applicable as probiotics due to Food regulation raw in Korea. That’s why we selected P. pentosaceus JBCC106 for further study.
Q2) The authors used cinnamoyl esterase producing bacteria as an entry point, but why did they not test the ability of CA production in the later application of fermenting broccoli juice.
After 36 h of fermentation of broccoli juice, the number of viable bacteria dropped sharply (figure 4A). If we put aside the fact that the antioxidant capacity of broccoli juice increased after fermentation (because the content of the antioxidant substance: sulforaphane increased significantly after fermentation of broccoli juice), does this application make sense? Since P. pentosaceus JBCC 106 has good antioxidant properties (table 1) and the fermentation of broccoli juice by this strain significantly increased the content of sulforaphane (an antioxidant substance, figure 5), why did the antioxidant capacity of broccoli juice not show a significant increase before and after fermentation (figure 4B), please The authors provide a reasonable explanation.
R2) Thanks for your opinion. There is two reason we fermented broccoli by using our selected strain P. pentosaceus JBCC106. Firstly, we want to know “fermentation ability” of our selected strain in broccoli matrix as model food system due to its high functional food material and simultaneously to know how many strains can be maintained their cells during fermentation. It would be desirable to provide high cell concentration as probiotics for human health. Secondly, we want to know whether our selected strain can be functional during fermentation or not. Actually, as your comments, we tried to detect CA production originally during fermentation, unfortunately we could not detect CA. We did not understand why this happened but we believed that we can detect CA if we optimize our fermentation condition for further in detail. However that is not our main focus on this study. Although we could not detect originally intended CA activity within broccoli matrix, we found dramatical increase of other functional materials like sulforaphane indicating our selected strain P. pentosaceus JBCC106 might be useful to prepare probiotics or postbiotics.
Concerning on antioxidant activity, We thought that the reason why total antioxidant level did not changed even if great increased sulforaphane level was might be decreased other antioxidant materials during long time fermentation. Due to the dramatically decreased living cells of P. pentosaceus JBCC106 after 11hrs fermentation as shown in Fig 4(A), might be affected almost identical level of antioxidants even greatly enhanced sulforaphane level. Actually we did not follow the change of sulforaphane during fermentation. Thus, it would be desirable to optimize fermentation condition if we want to know more exact reason and change of antioxidant activity and sulforaphane. However, our purpose in this study to find probiotic candidate stain which can be available for fermentation and to evaluate the probiotic properties of an isolate of P. pentosaceus JBCC 106 originating from the traditional Korean fermented food, Jangajji, and its viability and functionality in broccoli juice as model system. We are currently trying to optimize its fermentation condition.
Q3) In lines 227-229 (Table 2), P. pentosaceus JBCC 106 is less tolerant than LCC at pH 3, but significantly better than Lactobacillus rhamnosus at pH 2 and 1.5, please give a reasonable explanation.
R3) We are not sure why P. pentosaceus JBCC106 showed better acid tolerance at even lower pH compared to L. rhamnosus GG. But, it might be considered that the difference might be derived from P. pentosaceus JBCC106 origin which isolated from high salt added vegetable fermented food, Jangajji. However, We don’t know why P. pentosaceus JBCC106 did not showed better acid tolerance at pH 3 at this time. Sorry.
Q4) In line 162, I think it is more appropriate to use cell cytotoxicity than cell viability.
R4) We thank the reviewer for providing valuable comments and suggestions that have helped us improve the quality of our submitted manuscript. We changed the title of L162 to what you suggested and marked it in the manuscript of the revised manuscript following the reviewer’s comments.
Q5) Some writing errors, such as lines 170, 172, 241, 261, 268.
R5) Thanks for pointing out the error. We changed them. The corrections are marked in the manuscript of the revised manuscript following the reviewer’s comments.
Reviewer 3 Report
This paper tested the probiotic properties of a new Lactobacillus isolate Pediococcus pentosaceus, and prepared its application in the fermented food industry. There are specific comments as below:
1. There is a big disconnection in the writing of the abstract and introduction between the CA and the target strain Pediococcus pentosaceus. Please clarify the links between these two topics.
2. Please specify what does "NO" mean? Nitric Oxide?
3. It is not clear how the CE activities were defined. (lines 84-) And it would be useful to simultaneously detect the CA levels generated by the enzymatic process or the fermentation.
4. Line 101, please confirm the ABTS abs can be measured at 734 nm. Is this correct?
5. Please clarify how lactic acid fermentation is related to the core theme of the paper.
6. In the exp. design, it would be useful to add the positive and negative control strains throughout the paper to reflect the performances of Pediococcus pentosaceus.
There are some grammar mistakes.
Author Response
July 20, 2023
Microorganisms
Manuscript number: microorganisms-2497741
Title: Probiotic Properties of Pediococcus pentosaceus JBCC 106 and its Lactic Acid Fermentation on Broccoli Juice
We thank the editor and all reviewers for their thorough and constructive comments. We have addressed each comment in detail below, and the changes made to the manuscript are highlighted in the text. We kindly request your reconsideration of our manuscript for publication in Microorganisms.
Reviewer 3
Q1) There is a big disconnection in the writing of the abstract and introduction between the CA and the target strain Pediococcus pentosaceus. Please clarify the links between these two topics.
R1) Thanks for your comments. Our purpose in this study did not focused on CA, but probiotic properties of Pediococcus pentosaceus JBCC106 although this strain selected as high CA producer. We only use CA as an selection parameter from Jangajji due to its usefulness. Thus we removed abundant explanation on CE activity in introduction for better understanding and connection with broccoli fermentation.
Q2) Please specify what does "NO" mean? Nitric Oxide?
R2) We changed it. Thanks
Q3) It is not clear how the CE activities were defined. (lines 84-) And it would be useful to simultaneously detect the CA levels generated by the enzymatic process or the fermentation.
R3) Concerning on CE activity, we added reference. Actually, as your comments, we tried to detect CA production originally during fermentation, unfortunately we could not detect CA. We did not understand why this happened but we believed that we can detect CA if we optimize our fermentation condition. Although we could not detect originally intended CA activity within broccoli matrix, we found dramatical increase of other functional materials like sulforaphane indicating our selected strain P. pentosaceus JBCC106 might be useful to prepare probiotics or postbiotics.
Q4) Line 101, please confirm the ABTS abs can be measured at 734 nm. Is this correct?
R4) ABTS can result from the reaction between ABTS and potassium persulfate with direct production of a blue/green ABTS radical dot+ chromophore with absorption maxima at wavelengths of 645, 734, and 815 nm, or more commonly with absorption maxima at 415 nm. That’s why we selected ABTS abs at 734 nm.
Q5) Please clarify how lactic acid fermentation is related to the core theme of the paper.
R5) Our purpose in this study to find probiotic candidate stain which can be available for fermentation and to evaluate the probiotic properties of an isolate of P. pentosaceus JBCC 106 originating from the traditional Korean fermented food, Jangajji, and its viability and functionality in broccoli juice as model system. We only use CA producing activity as an screening marker for finding functional lactic acid bacteria. Since abundant explanation on CE at first paragraph of introduction might be cause big trouble so that we removed them. By fermenting broccoli as food matrix, we could understanding our selected strain can be useful preparing fermented functional juice with probiotics.
Q6) In the exp. design, it would be useful to add the positive and negative control strains throughout the paper to reflect the performances of Pediococcus pentosaceus.
R6) Actually. But it was difficult to select the positive and negative control strains. That’s why we selected L. rhamnose GG which currently used broadly at industry. All the papers published in our labs usually use L. rhamnose GG as control strain. Thanks.
Round 2
Reviewer 2 Report
The authors have carefully answered and revised my questions. I recommend that this manuscript be accepted for publication